# Factors associated with insufficient awareness of breast cancer among women in Northern and Eastern China: a case–control study

Li-Yuan Liu,[1,2] Yong-Jiu Wang,[1,2] Fei Wang,[1,2] Li-Xiang Yu,[1,2] Yu-Juan Xiang,[1,2] Fei Zhou,[1,2] Liang Li,[1,2] Qiang Zhang,[1,2] Qin-Ye Fu,[1,2] Zhong-Bing Ma,[1,2] De-Zong Gao,[1,2] Yu-Yang Li,[1] Zhi-Gang Yu[1,2]

[1]Department of Breast Surgery, The Second Hospital of Shandong University, Jinan, Shandong, China
[2]Institute of Translational Medicine of Breast Disease Prevention and Treatment, Shandong University, Jinan, Shandong, China

**Correspondence to**
Professor Zhi-Gang Yu; yzg@medmail.com.cn

## ABSTRACT

**Objectives** To investigate the awareness and knowledge level of breast cancer among Chinese participants.

**Design** Case–control study.

**Settings** This study was based on the database of the minister-affiliated hospital key project of the Ministry of Health of the People's Republic of China that included 21 Chinese hospitals between April 2012 and April 2013.

**Participants** Matched study was designed among 2978 participants with Han ethnicity aged between 25 and 70.

**Primary and secondary outcome measures** Student's t-test, Pearson's $\chi^2$ test, reliability analysis, exploratory factor analysis, and univariate and multivariate logistic regression analyses were performed to know the level of breast cancer knowledge and find the breast cancer awareness-associated factors.

**Results** 80.0% (2383/2978) of the participants had poor awareness level of breast cancer. In-depth knowledge of breast cancer such as early symptoms and risk factors was poorly found among them. Television broadcast and relatives or friends with breast cancers were the main sources of information about breast cancer. Of all participants, 72.8% (2167/2978) had heard about breast cancer as a frequent cancer affecting women, and 63.3% (1884/2978) knew that family history of breast cancer was a risk factor for breast cancer. Over half of them were aware that a breast lump could be a symptom of breast cancer. Multivariate analysis identified the following variables that predicted awareness of breast cancer: young age (OR=0.843, 95% CI 0.740 to 0.961), occupation (agricultural worker) (OR=12.831, 95% CI 6.998 to 23.523), high household social status (OR=0.644, 95% CI 0.531 to 0.780), breast hyperplasia history (OR=1.684, 95% CI 1.273 to 2.228), high behavioural prevention score (OR=4.407, 95% CI 3.433 to 5.657).

**Conclusion** Most women were aware of breast cancer as a disease, but their in-depth knowledge of it was poor. More publicity and education programmes to increase breast cancer awareness are necessary and urgent, especially for the ageing women and agricultural workers.

## INTRODUCTION

Breast cancer (BC) is one of the most common cancers and the leading cause of cancer-related death among women

### Strengths and limitations of this study

► This was a multicentre, matched case–control study designed to investigate the breast cancer awareness of women in 21 hospitals in Northern and Eastern China.

► We designed this hospital-based study to investigate the level of knowledge of breast cancer in female patients with breast cancer. Meanwhile, we compared the results with our previous community-based study.

► Although we used the same questionnaire in these two studies, we still did not have uniform standards and methods for measuring knowledge evaluation. Hence a standard measurement of breast cancer-related knowledge should be developed.

worldwide.[1] Despite that the incidence of BC was low in China, it has been increasing much faster than globally recently on account of change of diet, lifestyle and unique one-child policy.[2 3] According to latest statistics from the National Central Cancer Registry, BC is the most frequently diagnosed cancer among Chinese women in all age and ethnic groups, accounting for nearly one-fifth of all cancer types. Moreover, by 2011, BC incidence had increased to 32.43/100 000, which is higher than the average BC incidence in East Asia (27/100 000).[4]

Many studies have shown that the early detection of BC plays a vital role in patient survival. Further, a delay in the diagnosis and subsequent treatment can lead to worsening of morbidity and mortality.[5 6] From the time of onset to that of diagnosis, patients may experience disease progression, which could lead to tumour growth, and consequently, worse outcomes.[7] Stages at diagnosis differ among countries with different incomes. Salih *et al* reported that more than 70% of women

with BC in developed countries had disease stage I or II, compared with 20%–60% in low/middle-income countries.[8] A commercial report showed that nearly two-thirds of patients with BC were diagnosed with advanced disease in China, which was obviously higher than that in the USA.[2] It is well known that BC screening is an effective way to detect early-stage disease. However, the current nationwide BC screening programme is not available in China because of economic and demographic factors.[9] Additionally, people are not considered to consciously have the appropriate attitudes towards such BC screening programmes.[10] Many studies have shown that the level of cancer awareness is significantly related with early detection of BC.[10–12] Therefore, it is necessary to implement interventions aimed at increasing the comprehensive knowledge and awareness of BC symptoms and screening methods.

To promote BC awareness among Chinese women and build education programmes to prevent delays in diagnosis and treatment, healthcare specialists must know their current level of understanding. Thus, we performed this cross-sectional survey in China to assess the level of awareness and knowledge of BC-related symptoms and risk factors, and identify awareness-related factors.

## METHODS

### Study design

We designed a multicentre, case–control, hospital-based study to investigate the awareness of women in 21 hospitals located in 11 provinces in Northern and Eastern China. This study was conducted between April 2012 and April 2013 and was funded by the Ministry of Health of the People's Republic of China.

### Study population

All participants were of the Han ethnic group. Cases and controls were matched 1:1 for age (±3 years), diagnosis hospital (same hospital) and timing of examination (within 2 months). The inclusion criteria for BC group were as follows: (1) newly diagnosed and histologically confirmed BC and (2) women aged between 25 and 70 years old. Women who had recurrent or metastatic BC and/or complications of other malignant tumours confirmed by clinical or pathological diagnosis were excluded from this group. The members of control group were all from the regular physical examination centre in cooperation hospitals. The inclusion criteria for the control group were as follows: (1) negative physical examination results; (2) negative ultrasound scans of breast and/or mammographic screening results; and (3) no evidence or history of cancer. The exclusion criteria for the control group were as follows: a neoplastic disease at any other site, history of cancer or other major chronic diseases. We collected data strictly according to the inclusion and exclusion criteria, and then excluded subjects whose data were incomplete or lacking. Finally, a total of 1489 case–control sets were involved.

### Data collection

A self-designed structured questionnaire was previously developed to record information through person-to-person interviews. The theoretical basis of this interview questionnaire was numerous published articles and the opinions of a variety of experts in breast surgery, epidemiology, statistics, nutrition and molecular biology. Several similar questions were asked in different sections of the questionnaire to minimise recall bias. Previously, we conducted an investigation to assess the practicality and effectiveness of the survey and by which we validated the questionnaire.[9] The final interviewer-administered questionnaire was composed of the following parts: (1) demographic characteristics, physiological and reproductive factors, such as current age, age at menarche, age at menopause and menopausal status; (2) chronic diseases and family history such as breast hyperplasia, diabetes mellitus, hypertension and family history of BC; (3) lifestyle habits: smoking, alcohol intake, dietary habits and sleeping satisfaction; (4) awareness of BC-related knowledge: sources of BC-related information, risk factors and early signs or symptoms of BC (cumulative scores of these relevant items were counted as the related knowledge score and behavioural prevention score); (5) medical records including the visual examination, palpation and related diagnostic test results. Additionally, for patients with BC, the histological and immunohistochemical diagnoses were also evaluated.

### Scoring scheme

Awareness and knowledge of BC were assessed through 15 items on risk factors and early symptoms of BC included in the questionnaire (table 1). For each item, if respondents gave a correct response ('yes'), they scored 1 point; if a wrong response ('no' or 'do not know') was given, the score for the item was 0. Total scores thus ranged from 0 to 15. Then, we set a score to identify the status of respondents' awareness and knowledge of BC. Respondents with scores ranging from 0 to 8 were considered to have poor awareness and knowledge, whereas those with scores ranging from 9 to 15 points were considered to have high awareness and knowledge. Behavioural prevention was scored cumulatively by five items: participation in BC screening, breast self-examination (BSE), clinical breast examination (CBE), radiographic breast examination and breast ultrasound examination. Its scoring rules were the same as the 15-item questionnaire, and the total scores ranged from 0 to 5. The overall life satisfaction score was cumulatively based on 12 items; high scores meant low life satisfaction, and low scores indicated high life satisfaction.

### Quality control

Interviewers were selected by medical professionals and medical postgraduate students. All interviewers had completed standardised training and were certified to conduct surveys independently. The questionnaires and forms were coded twice and were entered twice by

**Table 1** Responses from the 15-item questionnaire for breast cancer awareness (n=2978)

| Questions | n | % |
|---|---|---|
| **Do you know breast cancer?** | | |
| Yes | 2167 | 72.8 |
| No | 736 | 24.7 |
| **Do you think screening is helpful for early detection of breast cancer?** | | |
| Yes | 2253 | 75.7 |
| No | 668 | 22.4 |
| **Do you think the early detection of breast cancer can improve survival?** | | |
| Yes | 2386 | 80.1 |
| No | 536 | 18.0 |
| *Knowledge about breast symptoms* | | |
| Local discomfort in breast | | |
| Yes | 1107 | 37.2 |
| No | 657 | 22.1 |
| Don't know | 1176 | 39.5 |
| Lump in breast | | |
| Yes | 1569 | 52.7 |
| No | 418 | 14.0 |
| Don't know | 953 | 32.0 |
| Axillary nodes | | |
| Yes | 1001 | 33.6 |
| No | 369 | 12.4 |
| Don't know | 1561 | 52.4 |
| Nipple retraction | | |
| Yes | 790 | 26.5 |
| No | 367 | 12.3 |
| Don't know | 1691 | 56.8 |
| Nipple discharge liquid | | |
| Yes | 875 | 29.4 |
| No | 367 | 12.3 |
| Don't know | 1691 | 56.8 |
| *Related factors of breast cancer* | | |
| Menarche at age before 12 | | |
| Yes | 381 | 12.8 |
| No | 471 | 15.8 |
| Don't know | 2088 | 70.1 |
| No parity or late childbirth | | |
| Yes | 679 | 22.8 |
| No | 377 | 12.7 |
| Don't know | 1880 | 63.1 |
| Menopause at a late age | | |
| Yes | 495 | 16.6 |
| No | 380 | 12.8 |

Continued

**Table 1** Continued

| Questions | n | % |
|---|---|---|
| Don't know | 2062 | 69.2 |
| Long-time drinking | | |
| Yes | 616 | 20.7 |
| No | 369 | 12.4 |
| Don't know | 1947 | 65.4 |
| High-fat diets | | |
| Yes | 636 | 21.4 |
| No | 353 | 11.9 |
| Don't know | 1947 | 65.4 |
| Long-term use of oestrogen drugs | | |
| Yes | 937 | 31.5 |
| No | 244 | 8.2 |
| Don't know | 1752 | 58.8 |
| Family history of breast cancer | | |
| Yes | 1884 | 63.3 |
| No | 158 | 5.3 |
| Don't know | 895 | 30.1 |
| Awareness of breast cancer | | |
| Highly aware | 595 | 20.0 |
| Poorly aware | 2383 | 80.0 |

different clerks. If there were inconsistent records, professionals would manually check and correct these. We also used computer software to check the logic and reasonable range of responses throughout the questionnaire to identify contradictory responses.

## Statistical analysis

EpiData V.3.1 was used to create the database. Statistical methods, including Student's t-test, Pearson's $\chi^2$ test, reliability analyses, exploratory factor analysis, and univariate and multivariate logistic regression analyses, were used to identify factors related to the knowledge of BC. ORs with 95% CIs were also calculated. All data analyses were performed using SPSS V.21.0.

## RESULTS

In this survey, 2978 women were included in our final analysis. The demographic characteristics are shown in table 2. The mean age of participants was 47.38±8.8 years. A total of 1611 subjects (54.1%) were from urban areas and 1248 (41.9%) were from rural areas. More than one-third of the population (33.9%) consisted of agricultural workers. Nearly half of the women (1422, 47.8%) had other occupations; some were teachers, drivers or provided civil service; 441 (14.8%) of them were workers; and 106 (3.6%) were medical staff. Regarding subject distribution based on education status, 466 subjects (15.6%) had primary school studies or less, 887

**Table 2** Main characteristics of responders (n=2978)

| Category | Awareness, n | | $\chi^2$ | P |
|---|---|---|---|---|
| | High | Low | | |
| **Present history of breast cancer** | | | | |
| Yes | 248 | 1241 | 20.585 | <0.001 |
| No | 347 | 1142 | | |
| **Age group (years)** | | | | |
| 25–34 | 56 | 149 | 18.655 | 0.001 |
| 35–44 | 214 | 723 | | |
| 45–54 | 206 | 950 | | |
| 55–64 | 108 | 490 | | |
| 65+ | 11 | 71 | | |
| Mean (SD) | 47.38 (8.8) | | | |
| **Location** | | | | |
| Urban | 465 | 1146 | 189.36 | <0.001 |
| Rural | 102 | 1146 | | |
| **Marital status** | | | | |
| Married | 559 | 2251 | 2.257 | 0.324 |
| Single | 10 | 23 | | |
| Widowed/divorced | 26 | 109 | | |
| **Monthly household income (¥)** | | | | |
| <1000 | 7 | 133 | 151.925 | <0.001 |
| 1000–1999 | 53 | 341 | | |
| 2000–2999 | 111 | 631 | | |
| 3000–4999 | 129 | 662 | | |
| ≥5000 | 281 | 540 | | |
| **Occupation** | | | | |
| Agricultural workers | 63 | 946 | 281.37 | <0.001 |
| Worker | 94 | 347 | | |
| Medical staff | 68 | 38 | | |
| Others | 370 | 1052 | | |
| **Educational status** | | | | |
| Primary or less | 26 | 440 | 186.918 | <0.001 |
| Middle | 122 | 765 | | |
| High | 216 | 698 | | |
| College | 201 | 364 | | |
| Postgraduate | 16 | 20 | | |
| **Menopause** | | | | |
| Yes | 175 | 760 | 1.520 | 0.218 |
| No | 400 | 1534 | | |
| **History of breast hyperplasia** | | | | |
| Yes | 200 | 388 | 91.447 | <0.001 |
| No | 383 | 1954 | | |

Continued

**Table 2** Continued

| Category | Awareness, n | | $\chi^2$ | P |
|---|---|---|---|---|
| | High | Low | | |
| **Family history of breast cancer** | | | | |
| Yes | 47 | 98 | 14.228 | <0.001 |
| No | 528 | 2178 | | |
| **Smoking** | | | | |
| Yes | 7 | 76 | 7.100 | 0.008 |
| No | 585 | 2298 | | |
| **Drinking** | | | | |
| Yes | 106 | 290 | 12.908 | <0.001 |
| No | 488 | 2080 | | |
| **BMI** | | | | |
| ≤23.9 | 290 | 1105 | 1.649 | 0.438 |
| 24.0–27.9 | 220 | 868 | | |
| ≥28.0 | 58 | 271 | | |
| **WHR** | | | | |
| <0.85 | 288 | 919 | 17.609 | <0.001 |
| ≥0.85 | 219 | 1063 | | |
| **Present life satisfaction** | | | | |
| High | 332 | 1492 | 9.308 | 0.002 |
| Low | 263 | 891 | | |
| **Behavioural prevention score** | | | | |
| Low | 263 | 891 | 9.308 | 0.002 |
| High | 332 | 1492 | | |

BMI, body mass index; WHR, waist-hip ratio.

(29.8%) had completed middle school, 914 (30.7%) had completed high school, 565 (19.0%) had college degrees and 36 (1.2%) had postgraduate degrees. Most of the responders (2810, 94.4%) were married, while 33 (1.1%) were single and 135 (4.5%) were widowed or divorced. In terms of income level, 140 (4.7%) earned less than ¥1000 per month, 394 (13.2%) earned ¥1000–¥1999 monthly, a quarter (24.9%) of the subjects had a monthly family income of ¥2000–¥2999, 791 (26.6%) of the women had monthly family income of ¥3000–¥4999 per month and 821 (27.6%) subjects had a monthly family income more than ¥5000.

Responses to questions about BC among the population are summarised in table 1. A total of 2167 subjects (72.8%) had heard about BC as a common disease affecting women, 2253 (75.7%) thought BC screening was helpful for early detection of BC and 2386 (80.1%) believed that early detection of BC could improve survival. In terms of the knowledge of risk factors for BC, 1884 (63.3%) knew that family history of BC was a risk factor for BC, and 381 (12.8%) knew that women whose menarche age was less than 12 years are at higher risk of developing BC; 679

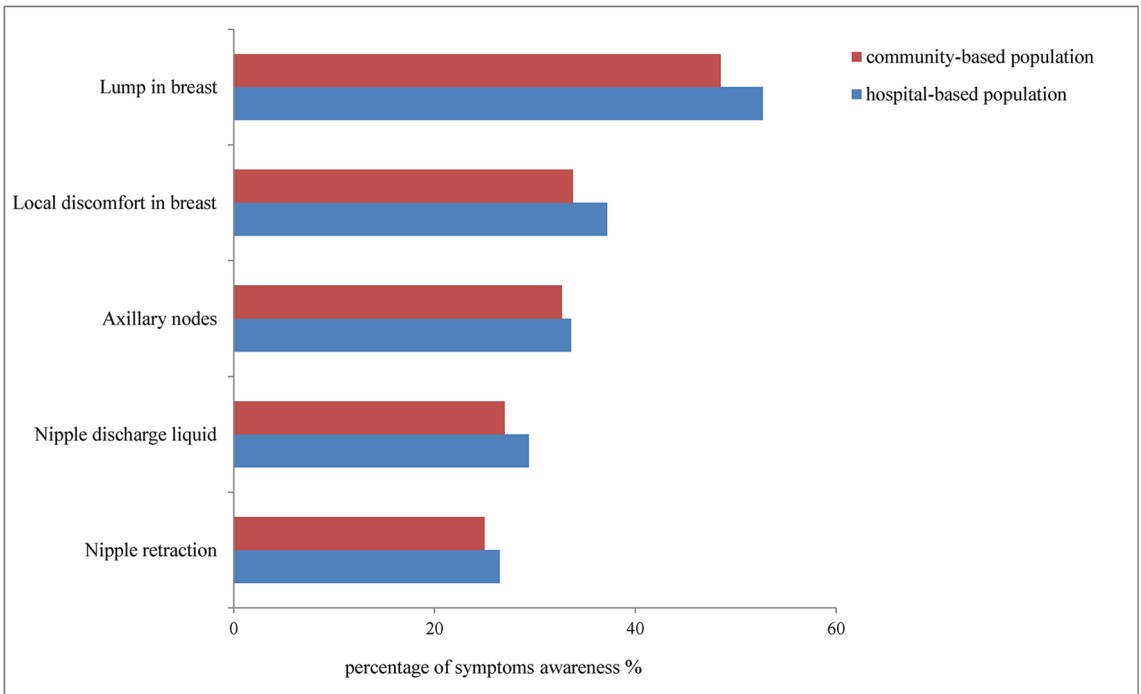

**Figure 1** Comparison of proportions of women aware of breast cancer symptoms between community-based and hospital-based studies.

(22.8%) knew that no parity and late childbirth increased the risk of BC, 495 (16.6%) knew that late menopause was associated with BC, 636 (21.4%) considered high-fat diet as a risk factor, 616 (20.7%) knew that long-term drinking habits increased BC risk and 937 (31.5%) knew that long-term use of oestrogen drugs was also a risk factor for BC. Regarding the awareness of early symptoms of BC, more than half (52.7%) of the subjects were aware that a breast lump could be a symptom of BC. Awareness of other symptoms was higher when compared with our previous community-based survey results (figure 1). In summary, 595 women (20.0%) showed high BC awareness and 2383 (80.0%) poor BC awareness.

Regarding the subjects' BC behavioural prevention, 1109 (37.2%) women had performed BSE at least once, 1020 of the 2978 women (37.2%) had undergone CBE, 35.9% had received breast ultrasound and 22.2% had undergone a radiographic breast examination. We set a 3-point cut-off value to distinguish high and low awareness of breast examination, and we found that more than half of the women (1824, 61.2%) had high awareness of behavioural prevention.

Most of the subjects obtained their information on BC from television (TV) broadcast and relatives or friends with BC, which accounted for 30.6% and 29.6% of all women, respectively (figure 2).

Results for univariate analysis indicated that awareness and knowledge of BC were related to history of BC, age, location, education status, occupation, household monthly income, social status, history of breast hyperplasia, family history of BC, drinking, smoking, physical training execution, waist-hip ratio, present life satisfaction and behavioural prevention score. All significant variables were included in multivariate analysis. Multivariate analysis identified that age (OR: 0.843, 95% CI 0.740 to 0.961), occupation (agricultural worker vs worker, medical staff: OR=3.066, 95% CI 1.999 to 4.703; OR=12.831, 95% CI 6.998 to 23.523, respectively), social status (OR: 0.644, 95% CI 0.531 to 0.780), history of breast hyperplasia (OR: 1.684, 95% CI 1.273 to 2.228) and behavioural prevention score (OR: 4.407, 95% CI 3.433 to 5.657) were independently correlated with BC awareness and knowledge (table 3) through stepwise method.

Reliability and construct validity and internal consistency reliability estimates of the 15-item scale of awareness and knowledge of BC were calculated using Cronbach's alpha. The α-coefficient for the total scale was 0.902, which was considered acceptable for internal consistency reliability.[13] Exploratory factor analysis was conducted to explore construct validity. The Kaiser-Meyer-Olkin (KMO) measure produced a coefficient of 0.883, indicative of excellent sampling adequacy. Bartlett's test of sphericity produced a value of 23825.328 (P<0.001), indicating that the correlation matrix was unlikely to be an identity matrix and was therefore suitable for factor analysis.[14]

## DISCUSSION
In this hospital-based study, we evaluated the level of BC awareness among Chinese women. Results showed that most participants had poor awareness regarding BC (80.0%), which is similar to our previous community-based study (81.4% of subjects had poor awareness). This seems to be a common phenomenon both in the low/middle-income countries and the developed

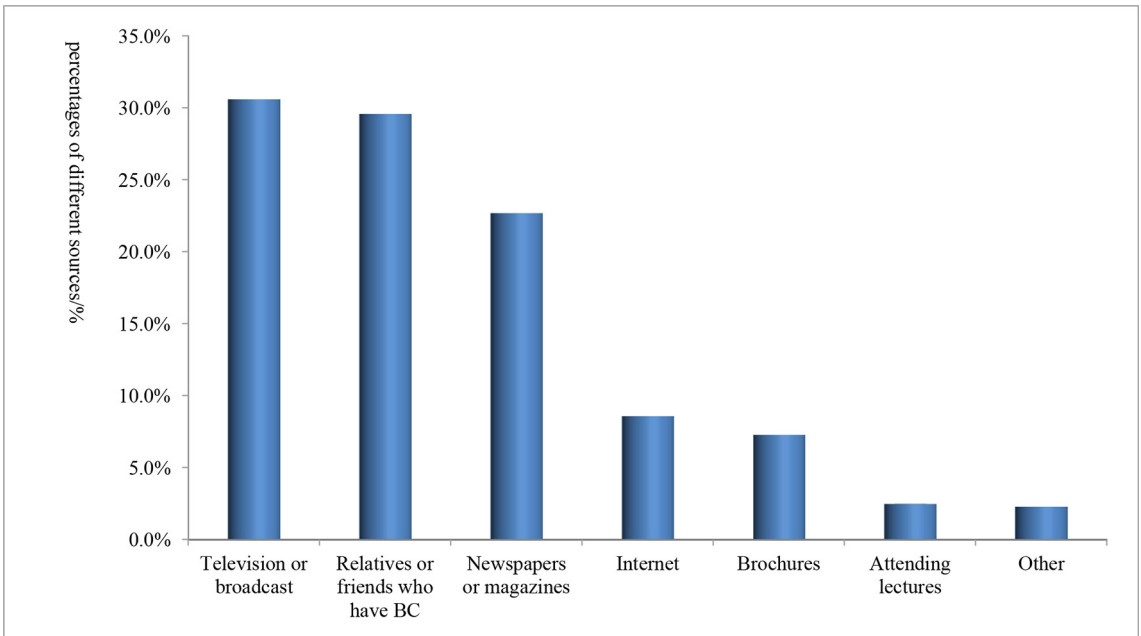

**Figure 2** Proportions of sources of information of breast cancer for women. BC, breast cancer.

countries, although the proportion of women with high awareness in developed countries was reportedly higher than that in low/middle-income countries or regions.[15 16]

It has also been published that different approaches for obtaining cancer-related knowledge influenced the level of awareness of BC.[17] We investigated the resources by which women obtained knowledge of BC, and we found that the majority of women obtained BC information through traditional media such as TV broadcasts (30.6%) and their friends or relatives with BC (29.6%). The internet has developed rapidly and become widespread, but in the present study, women failed to benefit from it in terms of BC awareness (8.6%). Less women participated in special lectures about BC (2.5%), which indicated that the efforts to publicise such events for BC

were likely insufficient. A study comparing non-Hispanic and Hispanic college women revealed that the internet was the most common information resource (75%),[18] while a Spanish survey indicated that the main sources of information were TV, press, family and friends.[19] In Malaysia and Cameroon, TV was still the major resource for obtaining knowledge.[20 21] It is worth noting that an increasing number of people are using the internet to obtain information on diseases such as cancer. However, in our country, people were not accustomed to employing the internet to search for cancer-related information, which may be a possible consequence of cultural and economic diversity.

In our study, more than two-thirds (72.8%) of the participants knew or had heard about BC, but their

| Table 3 | Multivariate analysis of factors related to knowledge of breast cancer | | | | | |
|---|---|---|---|---|---|---|
| | **B** | **SE** | **Wald** | **P** | **OR** | **95% CI** |
| Age | −0.170 | 0.067 | 6.511 | 0.011 | 0.843 | 0.740 to 0.961 |
| Occupation | | | | | | |
| Agricultural worker | | | | <0.001 | 1.000* | |
| Worker | 1.120 | 0.218 | 26.335 | <0.001 | 3.066 | 1.999 to 4.703 |
| Medical staff | 2.552 | 0.309 | 68.078 | <0.001 | 12.831 | 6.998 to 23.523 |
| Other | 1.238 | 0.183 | 45.721 | <0.001 | 3.448 | 2.408 to 4.936 |
| Household social status | −0.441 | 0.098 | 20.292 | <0.001 | 0.644 | 0.531 to 0.780 |
| History of breast hyperplasia | 0.521 | 0.143 | 13.298 | <0.001 | 1.684 | 1.273 to 2.228 |
| Behavioural prevention score | 1.483 | 0.127 | 135.509 | <0.001 | 4.407 | 3.433 to 5.657 |

Occupation: 'Other' includes occupations such as teacher, civil servant, individual business, driver, service, company employee and housewife.
Household social status, reference: low social status.
History of breast hyperplasia, reference: absence of breast hyperplasia history.
Behavioural prevention score was a cumulative score of 5 items; reference: low scores.
*Reference.

in-depth knowledge of the early symptoms of BC and risk factors was insufficient. Although more than half (52.7%) of them knew that the presence of a lump in the breast was a BC symptom and family history was an important risk factor for BC, the proportion of women who knew other BC symptoms and BC risk factors was low (overall, less than 40%). These results were consistent with our previous study[9] (figure 1), which indicated that although most women in China knew or had heard about BC, their in-depth knowledge of BC needed to be urgently improved. Studies from other countries showed consistent results,[5 15 22 23] especially those conducted in low/middle-income countries. In developed countries, the proportion of women reporting cancer-related symptoms was higher to a certain extent; this may be a possible consequence of their higher living standards, a greater consciousness of health and more social publicity.

Many studies found a close relationship between age and awareness and knowledge of BC.[10 22 24 25] The results of a study by Mandelblatt et al revealed that the level of knowledge of BC decreased as the age of responders increased.[26] Several articles researching Indian women also yielded similar results.[10 27] However, a study completed by Sen et al showed that older women were more interested in BC knowledge than younger women. Okobia et al also arrived at the same conclusion among Nigerian women.[28] In our study, we found that age was related to BC awareness as well (OR=0.843, 95% CI 0.740 to 0.961). Younger women tended to have more awareness and knowledge of BC although older women were at higher risk of BC development. We speculated that young women were more likely to focus on self-health conditions and be active learners and to access available information. From the above, it is urgent to improve the awareness and knowledge of older women to decrease the incidence of malignant breast tumours in this population.

The relationship between occupation and the level of awareness and knowledge of BC was also demonstrated in this study. Workers and medical staff tended to be more aware of the symptoms and risk factors for BC, while the awareness of agricultural workers was significantly poorer. Similar results were observed in many other studies including our community-based population survey.[9 29 30] Nonetheless, an Iranian study showed that the awareness of BC of rural women was moderate; this may indicate the existence of ethnic and population differences.[11] As shown in both community and hospital-based studies, medical personnel were more aware of BC than agricultural workers (OR: 4.774, 95% CI 4.316 to 5.281), likely benefiting from easier access to relevant knowledge.

Many studies demonstrated that education level was a major determinant of BC awareness and increased awareness was associated with higher levels of education. Terzioğlu et al[31] reported that the awareness of BC was attributed to the respondents' education level. This was also published in the study by Kotepui et al.[32] The univariate analysis in our study consistently showed that awareness and knowledge of BC were related to education

status; however, this result was not shown in our multivariate analysis. A study published in 2016[33] had a similar conclusion among Arabic Australian women, in which demographic characteristics, such as education level, were demonstrated as a negative factor. Education level was associated with several other factors such as occupation, income, among others, but further research is warranted to clarify other factors involved.

Women with a history of breast hyperplasia tended to know more about BC in our study (OR: 1.684, 95% CI 1.273 to 2.228). In some cases, the symptoms of breast hyperplasia may be similar to those of cancer; thus, women who had been diagnosed with breast hyperplasia were likely to pay more attention to breast health and to take better precautions. Breast hyperplasia may cause breast tenderness and other symptoms, but it may also serve as a forewarning for increased self-health awareness. This result was consistent with the research by Kosgeroglu et al,[34] in which it was shown that if women were diagnosed previously with benign breast diseases, they would make efforts to obtain sufficient knowledge of BC.

Regarding behavioural prevention of BC, our study showed that 61.2% of women adequately and consciously performed breast examination. Among the women with high awareness of BC, most had accepted breast-related examinations and 61.2% of them obtained a high behavioural prevention score. However, the proportions of women who accepted BSE, CBE, breast ultrasound or breast radiograph were still low (less than 40%). Rates of BSE practices, clinical examination and radiographic screening were lower than those reported in the studies of Whitman et al[35] and Kwok et al[36] (55.3% and 90% had a mammogram in Chicago and Australia, respectively). The cause for this difference may be the lack of related knowledge and insufficient guidance of treatment. The relationship between early detection and BSE remains controversial,[9] but most scholars still believe that self-examination probably improves awareness and might play an important role in nationwide programmes for earlier stage detection in China.[37 38] From this point of view, breast examination is helpful for early diagnosis and decreased mortality.

We compared the results with our earlier study.[9] Results showed that the proportion of women with poor awareness of BC was lower than that of community-based sample (80.0% vs 81.4%). As to the level of BC awareness, we also found higher percentages in hospital-based women (figure 1). It can be implied that medical community played an important role on it. Some studies showed the significance of medical staff on increasing level of cancer awareness. High levels of awareness about common health issues such as BC were shown in Terzioğlu's study based on Turkish population, as a consequence of direct communication with professional doctors.[31] Further study on importance of medical community in China should be conducted. When compared with the advanced countries, the level of BC awareness in China was lower, which was caused by many reasons. Financial obstacle

was one of the important reasons.[8] In China, average household income level could not catch up with that in advanced countries in a short time. Although there are no major differences socially, culturally and economically, the heightened awareness might be due to better infrastructure, advanced technology and educational facilities available in the advanced regions.[39 40] Low/middle-income countries provided women with limited access to professional knowledge and affordable quality healthcare treatment. In the meantime, BC screening was neither cost-effective nor feasible.[41]

In this study, the result of the reliability index was 0.902 and the validity was 0.883, while the results in our previous study were 0.910 and 0.870, respectively. We used the same questionnaire in these two studies, but we still did not have uniform standards and methods for measuring BC-related knowledge evaluation. Thus, we consider that a standard measurement of BC-related knowledge should be developed, which will be a part of important content of our further study. Additionally, the reality and validity of our 5-item questionnaire for assessing levels of behavioural prevention was not good enough for this assessment (Cronbach's alpha 0.769 and KMO coefficient 0.780), which limited the results of the study in terms of BC practices.

## CONCLUSIONS

Combined with our previous study, we concluded that most women were aware of BC as a disease entity, but their in-depth knowledge of the disease was poor. BC awareness, increased publicity and education programmes are necessary and urgent, especially for older women and agricultural workers.

**Acknowledgements** We thank all of the subjects involved in the study for their participation.

**Contributors** ZGY and LYL conceived and designed the interviews. LXY, FW, LL, QZ, YYL, DZG and QYF performed the interviews. YJW, YJX and LYL analysed the data. FZ and ZBM prepared the tables and figures. LYL, YJW and FW wrote the paper. ZGY supplied suggestions.

**Funding** This research was funded by the minister-affiliated hospital key project of the Ministry of Health of the People's Republic of China (Establishment and improvement of high-risk populations screening and evaluation system for breast cancer), the project of the National Natural Science Foundation of China (No: 81602912) and the project of the Second Hospital of Shandong University (No: S2015010014).

**Competing interests** None declared.

**Patient consent** Obtained.

**Ethics approval** The study protocol and procedures were approved by the Institutional Review Board at the Second Hospital of Shandong University. Before conducting the interviews and under the investigators' guidance, all participants provided written informed consent.

**Provenance and peer review** Not commissioned; externally peer reviewed.

**Data sharing statement** Anonymous data sets are available from the corresponding author (yzg@medmail.com.cn).

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
