## [Reviewer comments · BMJ Open]

ARTICLE DETAILS

TITLE (PROVISIONAL)	Factors Associated with Insufficient Awareness of Breast Cancer among Women in Northern and Eastern China : a case-control study
AUTHORS	Liu, Li-Yuan; Wang, Yong-Jiu; Wang, Fei; Yu, Li-Xiang; Xiang, Yu-Juan; Zhou, Fei; Li, Liang; Zhang, Qiang; Fu, Qin-Ye; Ma, Zhong-Bing; Gao, De-Zong; Li, Yu-Yang; Yu, Zhi-Gang

VERSION 1 – REVIEW

REVIEWER	Dr. Elima Jedy-Agba London School of Hygiene and Tropical Medicine, United Kingdom
REVIEW RETURNED	03-Aug-2017

GENERAL COMMENTS	Summary of Manuscript This study was carried out to investigate the awareness and level of knowledge of breast cancer among Chinese women. It was a case control study and the database of the Ministry of Health of the People's Republic of China that included 21 hospitals in China was used. Time period was between April 2012 and April 2013. In addition to exploratory factor analysis, the authors used multivariate logistic regression to find associations between various factors and breast cancer knowledge. 80% of women in the study had poor knowledge of breast cancer and its symptoms. The authors further identified occupation, household social status, history of breast hyperplasia and behavioral prevention score as factors that predicted knowledge of breast cancer in their study population. This study conducted is not novel and does not contribute any new evidence to the literature in this area. Abstract Participants: Line 21 should be 'Han ethnicity or ethnic group' not 'Han ethnic' In the results- the authors state that ...'80% of Chinese women had poor awareness of breast cancer' then subsequently in the conclusion they state that 'most women were aware of breast cancer as a disease. Although in-depth knowledge was poor. These 2 statements are contradictory and should be clarified. In the conclusion section, the authors state that....' Increased publicity and education programs to increase breast cancer awareness are necessary and urgent, especially targeting aging women and agricultural workers'. There were no results presented in the abstract to enable the authors arrive at this conclusion. Neither age nor being an agricultural worker was mentioned in the result as being associated with poorer knowledge.
---

	In addition, more clarity is needed in the presentation of results. For instance, Occupation is mentioned as being associated with breast cancer knowledge.... What occupation in this instance? For household social status, is it a poor social status or a high one that is associated with breast cancer knowledge? Was there an association found with age? Associations should be clearly defined as well as the direction of the effect. Introduction Line 9 – The authors state that, ‘...Although in the past, China had a relatively low incidence of BC, recently BC incidence has been increasing much faster in China than globally.’ Why is this so, the authors should offer some explanation as to why the BC incidence in China is rising faster than globally? Any risk factors particularly responsible for this increase in this population? Methods Study Population- Why was this age group chosen? women between 25 and 75 years? Why were women between 18-25 years excluded? A reason should be given for this. Is the incidence of BC extremely low in this age group in China? Data Collection- Line 21 should be ‘theoretical basis not bases’. The authors should explain how variables were selected for inclusion in the multivariate analysis. Were all the variables that were significant in the univariate analysis included in the multivariate? Results On page 8 line 24 The authors should include (SD) - it should be written as ‘The mean age (SD) of participants was.....’ not the mean age of participants was 47.38+ 8.8 years. Discussion: In the discussion section, the authors mentioned that in their study, young women tended to have more awareness than older women, the authors should discuss more on reasons why this might be so in this population. Page 20 line 12 the authors state that... ‘The univariate analysis in our study consistently showed that awareness and knowledge of BC were related to education status; however, the result was not shown in our multivariate analysis.’ The authors should explain why they think the effect of education was no longer significant in the multivariate analyses. The authors should reference this statement...’ The relationship between early detection and BSE remains controversial, but most scholars still believe that self-examination probably improves awareness and might play an important role in nationwide programs for earlier-stage detection in China.’ A reference should be provided to support this statement...’From this point of view, breast examination is helpful for early diagnosis and decreased mortality.’
--	--

REVIEWER	Marimer Santiago-Rivas Icahn School of Medicine at Mount Sinai (NY, US)
REVIEW RETURNED	28-Sep-2017

GENERAL COMMENTS	Summary: This is an analysis of data from a multicenter, case control hospital-based study to assess breast cancer awareness and knowledge in women of Han origin in Northern and Eastern China. The study identifies factors related to knowledge regarding breast cancer related information and behavioral prevention. Comments  1. The manuscript is well written, and the topic is relevant. 2. I am not sure what this study provides that is unique, when a similar study was already published by the authors. It is not clear what is the need for this manuscript. It seems the difference is that this study is a case-control study, and the previous one is a cross-sectional study (and a community-based sample). The results are similar, too. It would be important for the authors to clarify what unique information we get from this additional study. 3. The author should add information regarding participant recruitment. I have an idea how breast cancer patients were recruited, but I would like to have more information about how the control group was recruited. 4. In page 7, how did the authors set the score to determine high vs. poor awareness? 5. In the Discussion section, the authors should add details about the role of the medical community (e.g., doctors) for increasing breast cancer knowledge in Chinese women, and how could that be different from the US or other advanced countries.
---

VERSION 1 – AUTHOR RESPONSE

Reviewer #1

Please state any competing interests or state 'None declared':

Response: We are grateful for your hard and excellent effort on the evaluation of our manuscript. We state the competing interests in page 23 after the funding part.

Reviewer #2

1. The manuscript is well written, and the topic is relevant.

Response: Thank you very much for your high evaluation of our manuscript.

2. I am not sure what this study provides that is unique, when a similar study was already published by the authors. It is not clear what is the need for this manuscript. It seems the difference is that this study is a case-control study, and the previous one is a cross-sectional study (and a community-based sample). The results are similar, too. It would be important for the authors to clarify what unique information we get from this additional study.

Response: Thanks so much for your useful comments and we are very sorry that we did not differ this manuscript from the earlier article in detail. Our two articles both focused on the breast cancer awareness of eastern Chinese women, with regard to the differences, as you said, the previous article was a cross-sectional study based on the community women, and this article focused on the hospital-based study with case-control way. The purpose of this study included that to find out the differences of risk factors of breast cancer awareness and level of breast cancer-related knowledge between community-based and hospital-based sample. Generally, we found similar results that the awareness and knowledge among Chinese women were still low, but we also discovered that there are some differences. Such as, social status and history of hyperplasia were found to be risk factors only in this study, and previous factors like smoking and drinking did not appear. Additionally, this time we concluded the way women got information of breast cancer which was not referred in our previous article. So compared to our previous study, this manuscript was needed and some results were unique. We take the results comparison into our revised version in Discussion part in page 21.

3. The author should add information regarding participant recruitment. I have an idea how breast cancer patients were recruited, but I would like to have more information about how the control group was recruited.

Response: We are grateful for your hard and excellent effort on the evaluation of our manuscript. Our participants were selected from an epidemiological survey, funded by the Ministry of Health of the People's Republic of China, in 21 hospitals located in 11 provinces in Northern and Eastern China. The target population covered women aged 25–70 years of the Han ethnic group, and we then randomly selected the breast cancer group including 1489 patients in hospital and then randomly selected the control group by age (± 3 years), diagnosis hospital (same hospital) and timing of examination (within 2 months) matched, and the members of control group were all from the regular physical examination center in cooperation hospitals. Women who met the study requirements were then selected for the survey. We add the detail into our manuscript in study population part in page 5.

4. In page 7, how did the authors set the score to determine high vs. poor awareness?

Response: Thanks for your comments on our paper. We used the same questionnaire and scoring scheme to investigate awareness as our previous study. In our published article, the reliability and validity was conducted to be ideal, so we continue to use the 15-item scale and the cut-off value. To be honest, we still did not have uniform standards and methods for measuring breast cancer-related knowledge evaluation. Consequently, as we said in our manuscript, we consider that a standard measurement of breast cancer-related knowledge should be developed, which will be a part of important content of our further study. We add the explanation t in page 22.

5. In the Discussion section, the authors should add details about the role of the medical community (e.g., doctors) for increasing breast cancer knowledge in Chinese women, and how could that be different from the US or other advanced countries.

Response: Thanks so much for your comments. It can be implied that medical community played an important role on it. Some studies showed the significance of medical staff on increasing level of cancer awareness. High levels of awareness about common health issues such as breast cancer was showed in Gökay Terzioğlu's study based on Turkish population, as a consequence of direct communication with professional doctors. Further study on importance of medical community in China should be conducted. When compared with the advanced countries, the level of BC awareness in China was still lower, which was caused by many reasons. Financial obstacle was important one of reasons. In China, average household income level could not catch up with that in advanced countries in a short time. Although there are no major differences socially, culturally, and economically, the heightened awareness might be due to better infrastructure, advanced technology, and educational facilities available in the advanced regions. Developing countries provided women with limited access to professional knowledge and affordable quality healthcare treatment 18. Meantime, breast cancer screening was neither cost-effective nor feasible. We have revised the related Discussion part of our manuscript in page 21-22.

VERSION 2 – REVIEW

REVIEWER	Marimer Santiago-Rivas Icahn School of Medicine at Mount Sinai, USA
REVIEW RETURNED	17-Nov-2017
GENERAL COMMENTS	The authors addressed the issues stated in the previous review.

VERSION 2 – AUTHOR RESPONSE

Response to Reviewer Comments:

- Please revise the title so that it states the research question, study design, and setting. This is the preferred format for the journal.

Response: Thank you so much for your useful comments. According to your advice, we changed our title from “Hospital-based study of Breast Cancer Awareness among Women in Northern and Eastern China” to “Factors Associated with Insufficient Awareness of Breast Cancer among Women in Northern and Eastern China : a case-control study” .

- In the Results section of the abstract, please be clear that you are only looking at a sample (i.e., rather than saying "80.0% of Chinese women had poor awareness level of breast cancer" we recommend saying "80.0% of participants" and expressing the number as a fraction with the sample size as the denominator).

- Please report the 95% CI for ORs in the abstract

Response: Thanks for your comments on our paper. We have corrected the words “women” to “participants” in the abstract. Moreover, we also expressed the numbers behind the percentage and reported the 95% CI for ORs in the abstract according to your recommendation

- Please work to improve the quality of the English throughout your manuscript. We recommend asking a native English speaking colleague to assist you.

Response: Thank you very much for your high evaluation of our manuscript. We asked an American friend of PhD candidate to polish our paper. Some inappropriate expressions were corrected.